**The effects of environment on *Arctica islandica* shell formation and**
**architecture**
Stefania Milano[1*], Gernot Nehrke[2], Alan D. Wanamaker Jr.[3], Irene Ballesta-Artero[4,5], Thomas
Brey[2], Bernd R. Schöne[1]
[1] Institute of Geosciences, University of Mainz, Joh.-J.-Becherweg 21, 55128 Mainz, Germany
[2] Alfred Wegener Institute for Polar and Marine Research, Am Handelshafen 12, 27570 Bremerhaven,
Germany
[3] Department of Geological and Atmospheric Sciences, Iowa State University, Ames, Iowa, 50011-3212,
USA
[4] Royal Netherlands Institute for Sea Research and Utrecht University, PO Box 59, 1790 AB Den Burg,
Texel, The Netherlands
[5] Department of Animal Ecology, VU University Amsterdam, The Netherlands
* Corresponding author. Email: smilano@uni-mainz.de
**Keywords**: Confocal Raman microscopy; Scanning electron microscopy; Shell microstructure;
Water temperature; Diet; Bivalve shell

## Abstract

Mollusks record valuable information in their hard parts that reflect ambient environmental conditions. For this reason, shells can serve as excellent archives to reconstruct past climate and environmental variability. However, animal physiology and biomineralization, which are often poorly understood, can make the decoding of environmental signals a challenging task. Many of the routinely used shell-based proxies are sensitive to multiple different environmental and physiological variables. Therefore, the identification and interpretation of individual environmental signals (e.g. water temperature) often is particularly difficult. Additional proxies not influenced by multiple environmental variables or animal physiology would be a great asset in the field of paleoclimatology. The aim of this study is to investigate the potential use of structural properties of *Arctica islandica* shells as an environmental proxy. A total of eleven specimens were analyzed to study if changes of the microstructural organization of this marine bivalve are related to environmental conditions. In order to limit the interference of multiple parameters, the samples were cultured under controlled conditions. Three specimens presented here were grown at two different water temperatures (10 °C and 15 °C) for multiple weeks and exposed only to ambient food conditions. An additional eight specimens were reared under three different dietary regimes. Shell material was analyzed with two techniques: (1) Confocal Raman microscopy (CRM) was used to quantify changes of the orientation of microstructural units and pigment distribution and (2) Scanning electron microscopy (SEM) was used to detect changes in microstructural organization. Our results indicate that *A. islandica* microstructure is not sensitive to changes in the food source, and likely, shell pigment are not altered by diet. However, seawater temperature had a statistically significant effect on the orientation of the biomineral. Although additional work is

required, the results presented here suggest that the crystallographic orientation of biomineral units
of *A. islandica* may serve as an alternative and independent proxy for seawater temperature.


# 1. Introduction


Biomineralization is a process through which living organisms produce a protective, mineralized
hard tissue. The considerable diversity of biomineralizing species contributes to high variability in
terms of shape, organization and mineralogy of the structures produced (Lowenstam and Weiner,
1989; Carter et al., 2012). Different architectures at the micrometer and nanometer scale and
different biochemical compositions determine material properties that serve specific functions
(Weiner and Addadi, 1997; Currey, 1999; Merkel et al., 2007). Besides these differences, all
mineralized tissues are hybrid materials consisting in hierarchical arrangements of biomineral units
surrounded by organic matrix, also known as "microstructures" (Bøggild, 1930; Carter and Clark,
1985; Rodriguez-Navarro et al., 2006) or "ultrastructures" (Blackwell et al., 1977; Olson et al.,
2012) or overall "fabrics" (Schöne, 2013; Schöne et al., 2013). The carbonate and organic phases
represent the fundamental level of the organization of biomaterials (Aizenberg et al., 2005; Meyers
et al., 2006). The mechanisms of microstructure formation and shaping, especially in mollusks, has
attracted increasing attention during recent decades. At present, it is commonly accepted that the
organic compounds play an important role in the formation of the inorganic phases of biominerals
(Weiner and Addadi, 1991; Berman et al., 1993; Dauphin et al., 2003; Nudelman et al., 2006).
However, the identification of the exact mechanisms driving biomineralization is still an open
research question. Previous studies conducted on mollusks show that environmental parameters
can influence microstructure formation (Lutz, 1984; Tan Tiu and Prezant, 1987; Tan Tiu, 1988;
Nishida et al., 2012). These results set the stage for a research interest toward the use of shell
microstructures as proxies for reconstructing environmental conditions (Tan Tiu, 1988; Tan Tiu
and Prezant, 1989; Olson et al., 2012; Milano et al., 2015).

Mollusks are routinely used as climate and environmental proxy archives because they can

record a large amount of environmental information in their shells (Richardson, 2001; Wanamaker
et al., 2011a; Schöne and Gillikin, 2013). Whereas structures at nanometric level are still
underexplored as potential environmental recorders, shell patterns at lower magnification, such as
individual growth increments, are commonly used for this purpose (Jones, 1983; Schöne et al.,
2005; Marali and Schöne, 2015; Mette et al., 2016). Mollusks deposit skeletal material on a
periodic basis and at different rates (Thompson et al., 1980; Deith, 1985). During periods of fast
growth, growth increments are formed whereas during periods of slower growth, growth lines are
formed (Schöne, 2008; Schöne and Gillikin, 2013). The periodicity of such structures ranges from
tidal to annual (Gordon and Carriker, 1978; Schöne and Surge, 2012). By crossdating time-series
with similar growth patterns it is possible to construct century and millennia-long master
chronologies (Marchitto et al., 2000; Black et al. 2008; Black et al., 2016; Butler et al., 2013). This
basic approach, in combination with geochemical methods, has great potential in reconstructing
past climatic conditions (Wanamaker et al., 2011b). At present, the most frequently used and well-
accepted geochemical proxy is oxygen isotopic composition of shell material ($\delta^{18}O_{shell}$) (Epstein,
1953; Grossman and Ku, 1986; Schöne et al., 2004; Wanamaker et al., 2007) which may serve as
a paleothermometer and/or paleosalinometer (Mook, 1971; Andrus, 2011); however, $\delta^{18}O_{shell}$ value
is influenced by both seawater temperature and the isotopic composition of seawater ($\delta^{18}O_{water}$;
related to salinity). Thus, $\delta^{18}O_{shell}$-based temperature reconstructions are particularly challenging
in habitats with fluctuating $\delta^{18}O_{water}$ conditions such as estuaries or restricted basins (Gillikin et al.,
2005). Because of the multiple impacts on $\delta^{18}O_{shell}$ values, there have been substantial efforts to
develop alternative techniques to reconstruct environmental variables from mollusk shells (Schöne
et al., 2010; Milano et al., 2017).
This study investigates the possibility using shell microstructure properties to serve as a new
environmental proxy. For this purpose, the effects of seawater temperature (grown at 10 °C and 15
°C) and dietary regime on the microstructural units of *Arctica islandica* cultured under controlled
conditions were analyzed and quantified. *A. islandica* was chosen as model species because of its
great potential in paleoclimatology and paleoceanography (see Schöne, 2013; Wanamaker et al.,
2016). Its extreme longevity of up to more than 500 years makes this species a highly suitable
archive for long-term paleoclimate and environmental reconstructions (Schöne et al., 2005;
Wanamaker et al., 2008; Wanamaker et al., 2012; Butler et al., 2013).



## 2. Materials and Methods
The analyses were conducted on eleven *A. islandica* shells. Three juvenile *A. islandica* shells,
sampled for the seawater temperature experiment, were collected alive on November 21, 2009
aboard the *F.V. Three of a Kind* off Jonesport, Maine USA (44° 26′ 9.829″N, 67° 26′ 18.045″W)
in 82 m water depth. From 2009 to 2011, all animals were kept in a flowing seawater laboratory at
the Darling Marine Center, Walpole, Maine, USA (see Beirne et al., 2012 for additional details).
In 2011, clams were grown at two different temperature regimes for 16 weeks (Table 1). At the
completion of the experiment, shells were estimated to be between 4 to 5 years old. Eight one-year
old juveniles were collected in July 2014 from Kiel Bay, Baltic Sea (54° 32′ N, 10° 42′ E; Fig. 1)
and kept alive in tanks at 7 °C for six months at the Alfred Wegener Institute for Polar and Marine
Research (AWI), Bremerhaven, Germany. During this time interval, the animals were fed with an
algal mix of *Nannochloropsis* sp., *Isochrysis galbana* and *Pavlova lutheri.* Then, they were
transferred to the Royal Netherlands Institute for Sea Research (NIOZ), Texel, The Netherlands,
and cultured in tanks at three different dietary conditions for 11 weeks (Table 1).


2.1 Seawater temperature experiment
The seawater temperature experiment started on 27 March 2011 and ended on 21 July 2011. Prior
to the start of the experiment the animals were marked with calcein. The staining leaves a clear
fluorescent marker in the shells that can be used to identify which shell material has formed prior
to and during the experiment. Initially, the animals were kept at $10.3 \pm 0.2$ °C for 48 days. Then,
they were briefly removed from the tanks and marked again. Subsequently, the clams were cultured
for 69 more days at $15.0 \pm 0.3$ °C. Ambient seawater was pumped from the adjacent Damariscotta
River estuary and adjusted to desired temperature. The salinity was measured with a Hydrolab®
MiniSonde. It ranged between $30.2 \pm 0.7$ and $30.7 \pm 0.7$, in the two experimental phases,
respectively. During the entire culture period, all clams were exposed to ambient food conditions.
At the end of the experiment the soft tissues were removed.


2.2 Food experiment
The food experiment was carried out from 9 February 2015 to 29 April 2015. The animals were
placed in aquaria inside a climate room at 9 °C. Water temperature in the tanks ranged between 8
and 10 °C. Water salinity was measured by using an ENDECO 102 refractometer and ranged
between 29.6 and 29.9 ± 0.1 in each aquarium. The 15-liter tanks were constantly supplied with
aerated water from the Wadden Sea. The clams were acclimated for three weeks before the start of
the experiment. Three dietary regimes were chosen. One treatment consisted of feeding the animals
with Microalgae Mix (food type 1), a ready-made solution of four marine microalgae (25 %
*Isochrysis,* 25 % *Tetraselmis,* 25 % *Thalassiosira,* 25 % *Nannochloropsis*) with a particle size
range of 2 - 30 µm. A second treatment was based on PhytoMaxx (food type 2), a solution of living
*Nannochloropsis* algae with 2 - 5 µm size range. A third treatment served as control, i.e., the
animals were not fed with any additional food. In treatments with food type 1 and 2, microalgae
were provided at the constant optimum concentration of $20 \times 10^6$ cells/liter (Winter, 1969). A
dispenser equipped with a timer was used to distribute the food five times per day. At the end of
the experiment the soft tissues were removed. A distinct dark line in the shells indicated the
transposition to the NIOZ aquaria and the associated stress. This line marks the beginning of the
tank experiment.


2.3 Sample preparation
The right valve of each specimen was cut into two 1.5 millimeter-thick sections along the axis of
maximum growth. For this purpose, a low speed precision saw (Buehler Isomet 1000) was used.
Given the small size and fragility of the juvenile shells used in the food experiment, the valves
were fully embedded in a block of Struers EpoFix (epoxy) and air-dried overnight prior the
sectioning. Sections of the clams used in the temperature experiment were embedded in epoxy after
the cutting. All samples were ground using a Buehler Metaserv 2000 machine equipped with
Buehler silicon carbide papers of different grit sizes (P320, P600, P1200, P2500). In addition, the
samples were manually ground with Buehler P4000 grit paper and polished with a Buehler diamond
polycrystalline suspension (3 µm). Sample surfaces were rinsed in demineralized water and air-
dried.  In the samples of the temperature experiment, the calcein marks were located under a
fluorescence light microscope (Zeiss Axio Imager.A1m microscope equipped with a Zeiss
HBO100 mercury lamp and filter set 38: excitation wavelength, ca. 450 - 500 nm; emission
wavelength, ca. 500 - 550 nm).


## 2.4 *A. islandica* shell organization
The shell of *A. islandica* consists of pure aragonite and it is divided in two major layers, an outer
(OSL) and the inner shell layer (ISL). The OSL is further subdivided in outer (oOSL) and inner
portion (iOSL) (Schöne, 2013). These layers are characterized by specific microstructures (Ropes
et al., 1984; Fig. 2). The oOSL largely consists of homogenous microstructure with granular aspect
(Schöne et al., 2013). This type of architecture is characterized by approximately equidimensional
units of about 5 µm in width. The unit shape tends to be irregular with a bulky aspect. The
organization lacks of specific structural arrangement typical of other type of microstructures such
as the crossed-lamellar and cross-acicular microstructures. The latter are the main component
characterizing the iOSL and ISL (Dunca et al., 2009). Here, elongated units are arranged with two
main dip directions, resulting in a relative oblique alignment. As shown in Fig. 2, the elongation of
the structures becomes more evident in the ISL.
The present study focuses on ventral margin of the shells. Analyses were carried out exclusively in
the OSL.

Similar to other mollusks, the shell of *A. islandica* contains pigment polyenes which are

obviously visible when using CRM (Hedegaard et al., 2006). Polyenes are organic compounds
containing single (C-C) and double (C=C) carbon-carbon bonds forming a polyenic chain. Their
distribution across the shell is not homogenous. The pigments are abundant in the oOSL whereas
they become scarce in the iOSL. Furthermore, an enrichment in polyenes has been observed in the
growth lines, potentially indicating their involvement in the biomineralization process (Stemmer
and Nehrke, 2014). However, the specific functions of these organic compounds have not been
disclosed yet (Hedegaard et al., 2006; Karampelas et al., 2009). Given the high phenotypic
variation in pigmentation among and within mollusk species, it has been proposed that coloration
does not have a primary function as adaptive tool (i.e. camouflage, warning signaling) as in other
animals (Seilacher, 1972; Evans et al., 2009). This, in turn, can indicate a certain degree of
influence of the environment on the pigments, in particular by diet (Hedegaard et al., 2006;
Soldatov et al., 2013). In the current study, the effect of different dietary regimes was tested in
order to explore the potential of polyenes as environmental proxy.


2.5 Confocal Raman microscopy and image processing
Shells were mapped with a WITec alpha 300 R (WITec GmbH, Germany) confocal Raman
microscope. Scans of 50 × 50 μm, 100 × 50 μm and 150 × 50 μm were performed using a
piezoelectric scanner table. All Raman measurements were carried out using a 488 nm diode laser.
A spectrometer (UHTS 300, WITec, Germany) was used with a 600 mm$^{-1}$ grating, a 500 nm blaze
and an integration time of 0.03 s. On each sample two to six scans were made, depending of the
thickness of the shell. For instance, in juvenile shells (food experiment), two scans of each sample
were made. On larger shells used in the temperature experiment, six maps were completed, i.e.,
two maps in the oOSL, two in the middle of the iOSL and two in the inner portion of the iOSL.
Each scan contained between 40,000 and 120,000 data points, depending on the map size. The
spatial resolution equaled 250 nm. Half of the maps were performed on the shell portion formed
before the experiments. The other half were made on the shell portion formed under experimental
conditions. In order to avoid areas affected by transplantation or marking stress, the scans were
located far off the calcein and stress lines. Raman maps on food experiment shells were performed
300 μm away from the stress line. In the shells from the temperature experiment, the scans were
made 1 mm away from the calcein mark.
Polarized Raman microscopy is known to provide comprehensive information about the
crystallographic properties of the materials (Hopkins and Farrow, 1985). The aragonite spectrum
is characterized by two lattice modes (translation mode $T_a$, 152cm$^{-1}$ and librational mode $L_a$,
206cm$^{-1}$) and the two internal modes (in-plane band $v_4$, 705cm$^{-1}$ and symmetric stretch $v_1$, 1085cm$^{-}$
$^1$). The ratio ($R_{v1/Ta}$) between peak intensities belonging to $v_1$ and $T_a$ is caused by different
crystallographic orientations of the aragonitic units (Hopkins and Farrow, 1985; Nehrke and Nouet,
2011). Within each scan, $R_{v1/Ta}$ was calculated for each data point. New spectral images were
generated using WITecProject software (version 4.1, WITec GmbH, Germany). These images were
then binarized by replacing all values above 2.5 with 1 and the others with 0. The orientation was
quantified by calculating the area formed by pixels of value 1 over the total scan area. The imaging
software Gwyddion (http://gwyddion.net/last checked: June 2016) was used for this purpose. The
results were expressed in percentage.

The Raman scans of the food experiment shells were analyzed to investigate the pigment

composition. Polyene peaks have definite positions in the spectrum according to the number of the

C-C and C=C bonds of the chain, which are specific for certain types of pigments. The two major

polyene peaks at $\sim$ 1130 ($R_1$) and 1520 cm$^{-1}$ ($R_4$) were identified by using the "multipeak fitting 2"

routine of IGOR Pro (version 7.00, WaveMetrics, USA). Their exact position was determined

adopting a Gaussian fitting function (Fig. 3). The number of single ($N_1$) and double carbon bonds

($N_4$) was calculated by applying the equations by Schaffer et al. (1991):

(1)     $N_1 = 476 (R_1 - 1,082)$

(2)     $N_4 = 830 (R_4 - 1,438)$

Spectral images of the $R_4$ band were used to locate the polyenes in the shell and measure the

thickness of the pigmented layer. The images were analyzed using the software Panopea (© Schöne

and Peinl). The thickness of the pigmented layer was calculated as distance between the outer shell

margin and the point where the concentration of polyenes suddenly declined. The measurements

were taken perpendicular to the shell outer margin. This analysis was conducted only on the shells

of the food experiment. Given the larger size of the shells used in the temperature experiment, the

spectral maps were not sufficient for a correct localization of the pigmented layer boundaries and

estimation of its thickness.

To quantify changes of the orientation of individual biomineral units of the juvenile shells

(food experiment), the spectral maps were subdivided into two portions. The outermost shell

portion (oOSL) was enriched in pigments whereas the iOSL showed a decrease in polyene content.

## 2.6 Scanning electron microscopy

After performing Raman measurements, the samples were prepared for SEM analysis. Each shell slab was ground with a Buehler Metaserv 2000 machine and Buehler silicon F2500 grit carbide paper. To reduce the impact of grinding on the sample surface of juvenile shells, extra grinding was done by hand. Then, the slabs were polished with a Buehler diamond polycrystalline suspension (3 μm). Afterward, shell surfaces were etched in 0.12 N HCl solution for 10 (food experiment samples) to 90 s (temperature experiment samples) and subsequently placed in 6 vol % NaClO solution for 30 min. After being rinsed in demineralized water, air-dried samples were sputter-coated with a 2 nm-thick platinum film by using a Low Vacuum Coater Leica EM ACE200.

A scanning electron microscope (LOT Quantum Design 2[nd] generation Phenom Pro desktop SEM) with backscattered electron detector and 10 kV accelerating voltage was used to analyze the shells. Images were taken at the same distances from the calcein and stress lines as in the case of the Raman measurements to assure comparability of the data.

In addition, stitched SEM images of the ventral margins were used to accurately determine the shell growth advance during the culturing experiments. Growth increment widths were measured with the software Panopea. Given the difference in duration of the two phases of the temperature experiment, the measurements were expressed as total growth and instantaneous growth rate (Fig. 4a + b). The latter was calculated using the following equation (Brey et al., 1990; Witbaard et al., 1997):

(3)  Instantaneous growth rate = $(\ln (y_t / y_0) / a)$

where $y_0$ represents the initial shell height, $y_t$ is the final shell height and $a$ is the duration of the experiment. In the case of the food experiment, only the total growth was calculated (Fig. 4c).

## 3. Results

### 3.1 Effect of seawater temperature and diet on *A. islandica* shell growth

When exposed to a water temperature of 10 °C, the shells grew between 11.67 and 14.17 mm during a period of 48 days. During a period of 69 days at 15 °C, the growth ranged between 2.32 and 5.77 mm (Fig. 4a). Instantaneous growth rate showed a decrease between the two experimental phases. At 10 and 15 °C, the average instantaneous growth per day was 0.0091 and 0.0013, respectively (Fig. 4b). The decrease in total growth and growth rate between the two temperatures was statistically significant (*t*-test, $p < 0.01$).

During the food experiment, shells grew between 0.37 and 3.71 mm with large differences due to the different food types. Growth of specimens exposed to food type 1 ranged between 1.87 to 3.71 mm, whereas those cultured with food type 2 grew between 0.55 to 0.96 mm. Both control specimens added 0.37 mm of shell during the experimental phase (Fig. 4c). ANOVA and Tukey´s HSD post hoc tests showed significant differences between specimens cultured with food type 1 and 2 ($p < 0.05$) and between food type 1 and control shells ($p < 0.05$).



### 3.2 Effect of seawater temperature on *A. islandica* microstructure

At a water temperature of 10 °C, the area occupied by microstructural units oriented with $R_{v1/Ta}$
higher than 2.5 a.u. (= arbitrary units) ranged between 31.3 and 50.6 % in the oOSL and between
21.3 and 33.5 % in the iOSL. When exposed to 15 °C, values ranged between 25.6 and 48.7 % and
between 45.7 and 55.9 % in the oOSL and iOSL, respectively (Fig. 5). Whereas the slight difference
of area with $R_{v1/Ta} > 2.5$ in the oOSL was not significant between the two water temperatures (*t*-
test, $p = 0.62$), the area with $R_{v1/Ta} > 2.5$ in the iOSL significantly increased at 15 °C (*t*-test, $p =$
0.02). Under the SEM, no difference was visible between units formed at 10 °C and 15 °C (Fig. 6).


## 3.3 Effect of food on *A. islandica* microstructure and pigments
In the shells cultured with food type 1, the area occupied by biomineral units oriented with $R_{v1/Ta}$
higher than 2.5 a.u. during the experiment ranged between 24.8 % (oOSL) and 43.0 % (iOSL). In
the shell portion deposited during the acclimation phase, the ratio varied between 19.4 % (oOSL)
and 36.2 % (iOSL). Although a trend was recognized, these variations were not statistically
different (*t*-tests. OSL: $p = 0.43$; ISL: $p = 0.57$; Fig. 7a). On the contrary, in the clams exposed to
food type 2, the area occupied by units oriented with $R_{v1/Ta} > 2.5$ ranged between 11.7 % (oOSL)
and 20.4 % (iOSL). Before the experiment, the proportions were higher, i.e., 18.1% (oOSL) and
26.3% (iOSL) (Fig. 7b). As for the other treatment, the difference was not significant (*t*-tests.
oOSL: $p = 0.34$; iOSL: $p = 0.28$). In the control shells grown with no extra food supply, the area
with $R_{v1/Ta} > 2.5$ ranged between 24.6 % (oOSL) and 44.8 % (iOSL) during the experiment and
21.2 % (oOSL) and 44.5 % (iOSL) before the experiment (Fig. 7c). Hence, no trend was visible
and the two portions did not show significant differences (*t*-tests. oOSL: $p = 0.59$; iOSL: $p = 0.99$).
As for the temperature experiment, under the SEM, the microstructure of the shells from the food
experiment did not show any change (Fig. 9).

All treatments showed a slightly thicker pigmented layer formed during the experiment than

during the acclimation phase (Fig. 9a). During the experiment, clams cultured with food type 1
showed, on average, a thickening by 6.4 %. In the food type 2 specimens, the layer thickness
increased by 9.9 %. Control shells showed an increase of 10.4 % (Fig. 9b). However, none of these
differences was statistically significant ($t$-test. Food type 1: $p = 0.43$; Food type 2: $p = 0.39$; Control:
$p = 0.10$). According to the position of the polyene peaks, the number of single carbon bonds in the
pigment chain did not change between the acclimation and experimental phase ($N_1 = 10.1 \pm 1.3$
and $N_1 = 10.0 \pm 0.9$, respectively). Likely, no significant variation was observed in the number of
double carbon bonds ($N_4 = 10.5 \pm 0.2$ and $N_4 = 10.4 \pm 0.3$, respectively; Table 2).




# 4. Discussion
According to the results, variations of both food type and water temperature can influence the shell
production rate of *A. islandica*. However, the shell microstructure and pigmentation react
differently to these two environmental variables. Whereas changes of the dietary conditions do not
affect the shell architecture and pigment composition, the crystallographic orientation of the
biomineral units responds to seawater temperature fluctuations.


## 4.1 Environmental influence on shell microstructure

The environmental conditions experienced by mollusks during the process of biomineralization appear to influence shell organization (Carter, 1980). Among the different environmental variables, water temperature is the most studied driving force of structural changes of the shell. For instance, shell mineralogy can vary depending on water temperature (Carter, 1980). According to the thermal potentiation hypothesis, nucleation and growth of calcitic structural units is favored at low temperatures by kinetic factors (Carter et al., 1998). As a consequence, bivalve species living in cold water environments exhibit additional or thicker calcitic layers compared to the corresponding species from warm waters (Lowenstam, 1954; Taylor and Kennedy, 1969). Changes in the calcium carbonate polymorph also affect the type of microstructures (Milano et al., 2016). However, architectural variations often occur without mineralogical impact (Carter, 1980).

The present results indicate that temperature induces a change in the crystallographic orientation of the biomineral units of *A. islandica*. Although water temperature was previously shown to have an impact on microstructure formation, the attention has been mainly addressed to the effects on the morphometric characteristics (e.g. size and shape) or on the type of microstructure. Milano et al. (2017) demonstrated that size and elongation of prismatic structural units of *Cerastoderma edule* were positively correlated to seawater temperature variation throughout the growing season. Likely, low temperatures induced the formation of small nacre tablets in *Geukensia demissa* (Lutz, 1984). Seasonal changes of the microstructural type were reported in the freshwater bivalve *Corbicula fluminea* (Prezant and Tan Tiu, 1986; Tan Tiu and Prezant, 1989). During the warm months, crossed acicular structure was produced, whereas simple

crossed-lamellae were formed during the winter period. So far, variations of the crystallographic
properties of bivalve biominerals have been exclusively investigated as a response to hypercapnic
(acidified) conditions. *Mytilus galloprovincialis* and *Mytilus edulis* showed a significant change in
the orientation of the prisms forming shell calcitic layer when subjected to hypercapnia (Hahn et
al., 2012; Fitzer et al., 2014).  Altered crystallographic organization may derive from the animal
exposure to suboptimal conditions. These findings together with the present results suggest that
thermal- and hypercapnic-induced stress are likely to affect the ability of the bivalves to preserve
the orientation of their microstructural units (Fitzer et al., 2015).

Different food sources do not significantly influence the orientation of the biomineral units

or the composition and distribution of pigments in shells of *A. islandica*. In previous studies, the
relationship between microstructure and diet was virtually overlooked resulting in a lack of data in
the literature. As suggested by Hedegaard et al. (2006), however, the type of polyenes is influenced
by food. The ingestion of pigment-enriched microalgae potentially leads to an accumulation of
pigments in mollusk tissues and the shell (Soldatov et al., 2013). On the other hand, it has been
argued that polyenes do not generate from food sources like other pigments (i.e., carotenoids), but
they are locally synthesized (Karampelas et al., 2009). In accordance to Stemmer and Nehrke
(2014), the results presented here support the view that the specific diets on which the animals rely
on do not influence shell pigment composition. The chemical characteristics of the polyenes are
likely to be species-specific and independent from the habitats.


4.2 Confocal Raman microscopy as tool for microstructural analysis
From a methodological perspective, the present study represents an innovative approach in
the investigation of shell microstructural organization. Electron backscatter diffraction (EBSD) has
been previously used to determine the crystallographic orientation of gastropod (Fryda et al., 2009;
Pérez-Huerta et al., 2011) and bivalve microstructural units (Checa et al., 2006; Frenzel et al., 2012;
Karney et al., 2012). Whereas, CRM on mollusk shells is generally applied within studies on
taphonomic mineralogical alteration and pigment identification (Stemmer and Nehrke, 2014;
Beierlein et al., 2015). Both techniques provide considerably high spatially resolved analysis up to
250 nm, allowing the identification of individual structural units at μm- and nm-scale (Cusack et
al., 2008; Karney et al., 2012). CRM offers important advantages supporting a broader application
of this methodology in the biomineralization research field. For instance, samples do not require
any pre-treatment. Unlike EBSD, there is no need of preparing thin-sections (~ 150 μm thick) or
etching the shell surface (Griesshaber et al., 2010; Hahn et al., 2012). Therefore, further structural
and geochemical analyses can be easily performed on the same sections (Nehrke et al., 2012). In
addition, the size of CRM scans can be remarkably large (~ 7-8 mm$^2$) without compromising the
achievable resolution. By overlapping adjacent scans, it is possible to produce stitched scans
allowing to further increase the region of interest on the shell surface. On the other side, EBSD
provides a relevant advantage to take into consideration. It allows absolute measures of the
crystallographic orientation of the carbonate structures. The CRM, instead, determines the relative
change in the orientation between the single units without providing absolute values.
SEM has previously been demonstrated to provide a convenient approach for the
identification of individual structural units and the quantification of potential changes occurring
within them (Milano et al., 2017, 2016b). However, SEM exclusively provides information about
the morphometric characteristics of the microstructural units. As highlighted by the present study,
to achieve an exhaustive examination, it is suggested to combine SEM with techniques assessing
crystallographic properties of the biomaterials. For instance, our results show that the effect of
water temperature is detectable in crystallographic orientation but not in morphometric features of
the biomineral units.

## 4.3 Environmental influence on shell growth
Numerous previous studies demonstrated that growth rate of *A. islandica* is linked to environmental
variables (e.g., Witbaard et al., 1997, 1999; Schöne et al., 2004; Butler et al., 2010; Mette et al.,
2016). However, the relative importance of the main factors, temperature and food supply/quality
driving shell formation are still not well understood. Positive correlations between shell growth
and water temperature have been identified (i.e., Schöne et al., 2005; Wanamaker et al., 2009;
Marali et al., 2015), but the relationship between shell growth and environment is more complex
(Marchitto et al., 2010; Stott et al., 2010; Schöne et al., 2013) and likely dependent on the synergic
effect of food availability and water temperature (Butler et al., 2013; Lohmann and Schöne, 2013;
Mette et al., 2016). Tank experiments were run in order to precisely identify the role of these two
parameters of shell growth of *A. islandica* (Witbaard et al., 1997; Hiebenthal et al., 2012). A tenfold
increase in instantaneous growth rate was observed between 1 and 12 °C, with the greatest variation
occurring below 6 °C (Witbaard et al., 1997). On the contrary, a temperature increase between 4
and 16 °C was shown to produce a slowdown of shell production (Hiebenthal et al., 2012). Our
results are in agreement with the latter study and show a decrease in the instantaneous growth rate
between 10 and 15 °C. High temperatures are often associated with an increase of free radical
production (Abele et al., 2002). A large amount of energy then has to be allocated to limit oxidative
cellular damage (Abele and Puntarulo, 2004). This translates into a higher accumulation of
lipofuscin and slower shell production rate (Hiebenthal et al., 2013). The contrasting results of
previous studies may be explained by individual differences in the tolerance toward temperature
change (Marchitto et al., 2000).

Along with water temperature, food availability was also shown to influence *A. islandica*

shell growth (Witbaard et al., 1997). At high algal cell densities, the siphon activity increased. This,
in turn, was positively correlated to shell growth. Previous experiments used different combinations
of algae such as *Isochrysis galbana* and *Dunaliella marina* (Witbaard et al., 1997), or
*Nannochloropsis oculata, Phaeodactylum tricornutum* and *Chlorella* sp. (Hiebenthal et al., 2012)
to grow the clams. However, there are still uncertainties about the composition of the primary food
source for this species (Butler et al., 2010). Even though it is challenging to determine the preferred
algal species, our results show that the use of a mixture of different algal species results in
significantly faster shell growth than the used of just one algal species. In the natural environment,
suspension feeders such as *A. islandica* preferentially ingest certain particle sizes (Rubenstein and
Koehl, 1977; Jorgensen, 1996; Baker et al., 1998). The exposure to a limited algal size range, as in
the case of food type 2, may affect shell growth. Furthermore, multispecific solutions contain a
higher variability of biochemical components that better meet the nutritional requirements of the
animal (Widdows, 1991). Our results are in good agreement with previous findings. For instance,
it has been shown by Strömgren and Cary (1984) that *Mytilus edulis* shell growth increased as a
result of a diet based on three different algal species. Furthermore, Epifanio (1979) tested the
differences on the growth of *Crasssostrea virginica* and *Mercenaria mercenaria* of a mixed diet
composed by *Isochrysis galbana* and *Thalassiosira pseudonana* and diets consisting of the single
species. Faster growth was measured in the mixed diet treatment, indicating a synergic effect of
the relative food composition (Epifanio, 1979). Likely, *Mytilus edulis* grew faster when reared with
different types of mixed diets as opposed to monospecific diets (Galley et al., 2010).



## 5. Conclusions

*Arctica islandica* shell growth and biomineral orientation vary with changes in seawater temperature. However, exposure to different food sources affect shell deposition rate but do not influence the organization of the biomineral units. Given the exclusive sensitivity to one environmental variable, the orientation of biomineral units may represent a promising new temperature proxy for paleoenvironmental reconstructions. However, additional studies are needed to further explore the subject. In particular, intra-individual variability influence on the results needs to be assessed. In the present study, a variation in the orientation between individuals was well visible and the risks associated have to be taken in account when considering further application of the possible proxy. Furthermore, the effect of other environmental variables such as salinity needs to be tested.

The innovative application of CRM for microstructural orientation and proxy development proved that the technique has large potential in this research direction. More studies are needed to validate its suitability in paleoclimatology experimental works.



## Acknowledgements

The authors acknowledge the crew of the *F.V. Three of a Kind* for helping with the collection of
the animals. Design and execution of the seawater temperature experiment were successfully
realized thanks to the support of B. Beal, D. Gillikin, A. Lorrain and the Darling Marine Center
scientific team. Funding for this study was kindly provided by the EU within the framework of the
Marie Curie International Training Network ARAMACC (604802).

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

# Figures and tables

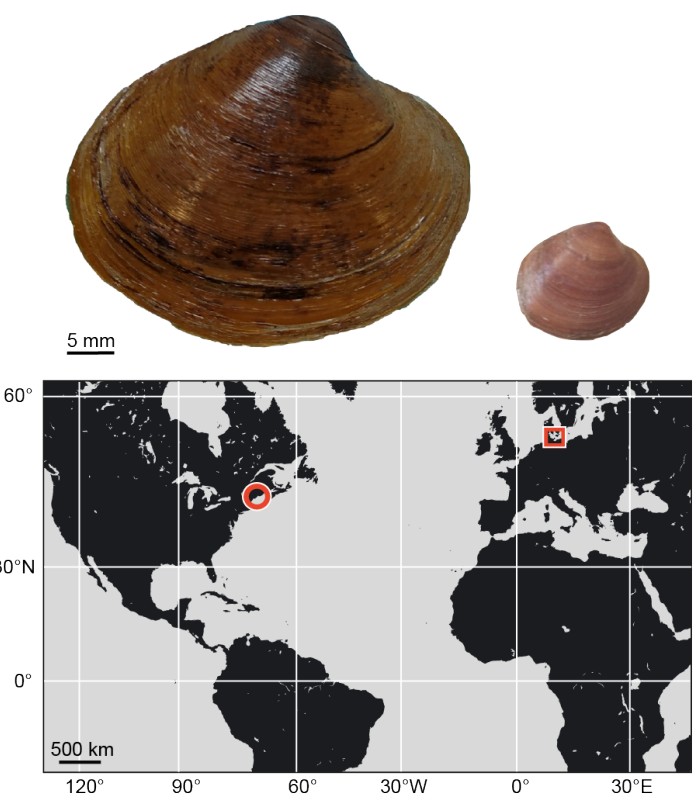


**Fig. 1.** Shell of adult *Arctica islandica* used in the temperature experiment (left) and juvenile from the Baltic Sea used in the food experiment (right). The map indicates the localities where the two sets of shells were collected: Jonesport, Maine (circle) and Kiel Bay (square).

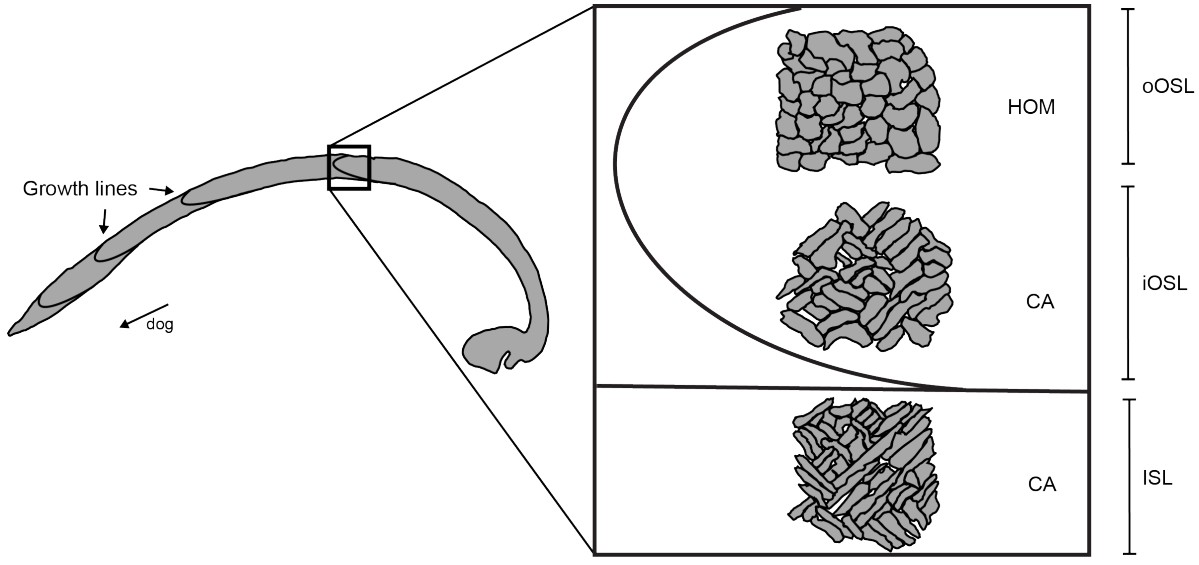


**Fig. 2.** Sketch showing the microstructures characterizing the different shell layers of *Arctica islandica.* The
oOSL is formed by homogenous microstructure (HOM), whereas the oOSL and ISL are composed of
crossed acicular structure (CA). dog = direction of growth.

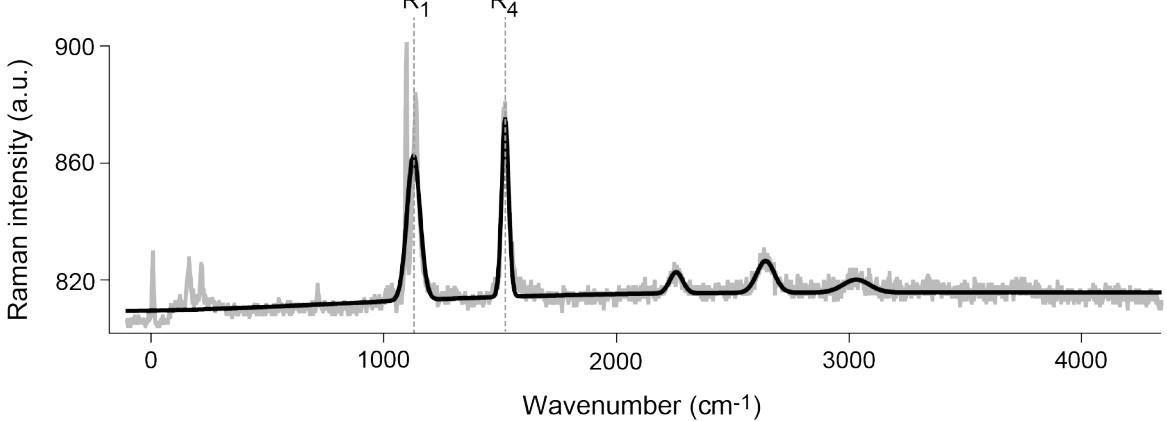


**Fig. 3.** Raman spectrum of *Arctica islandica* showing the typical aragonite peaks (grey line). The exact
position of the polyene peaks $R_1$ and $R_4$ was determined by using a peak fitting routine based on a Gaussian
function (black line).


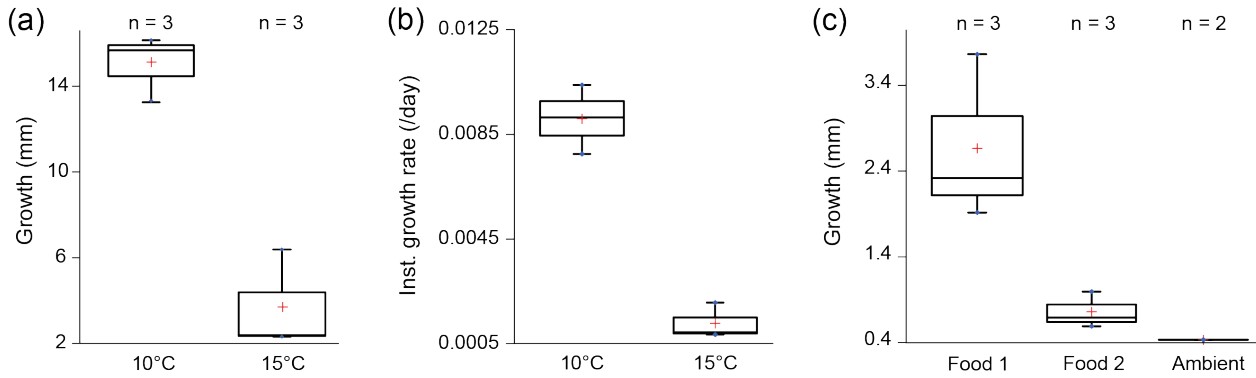


**Fig. 4.** *Arctica islandica* shell growth under controlled conditions. (a) Total growth and (b) instantaneous
growth rate during the temperature experiment. (c) Total growth during the food experiment.

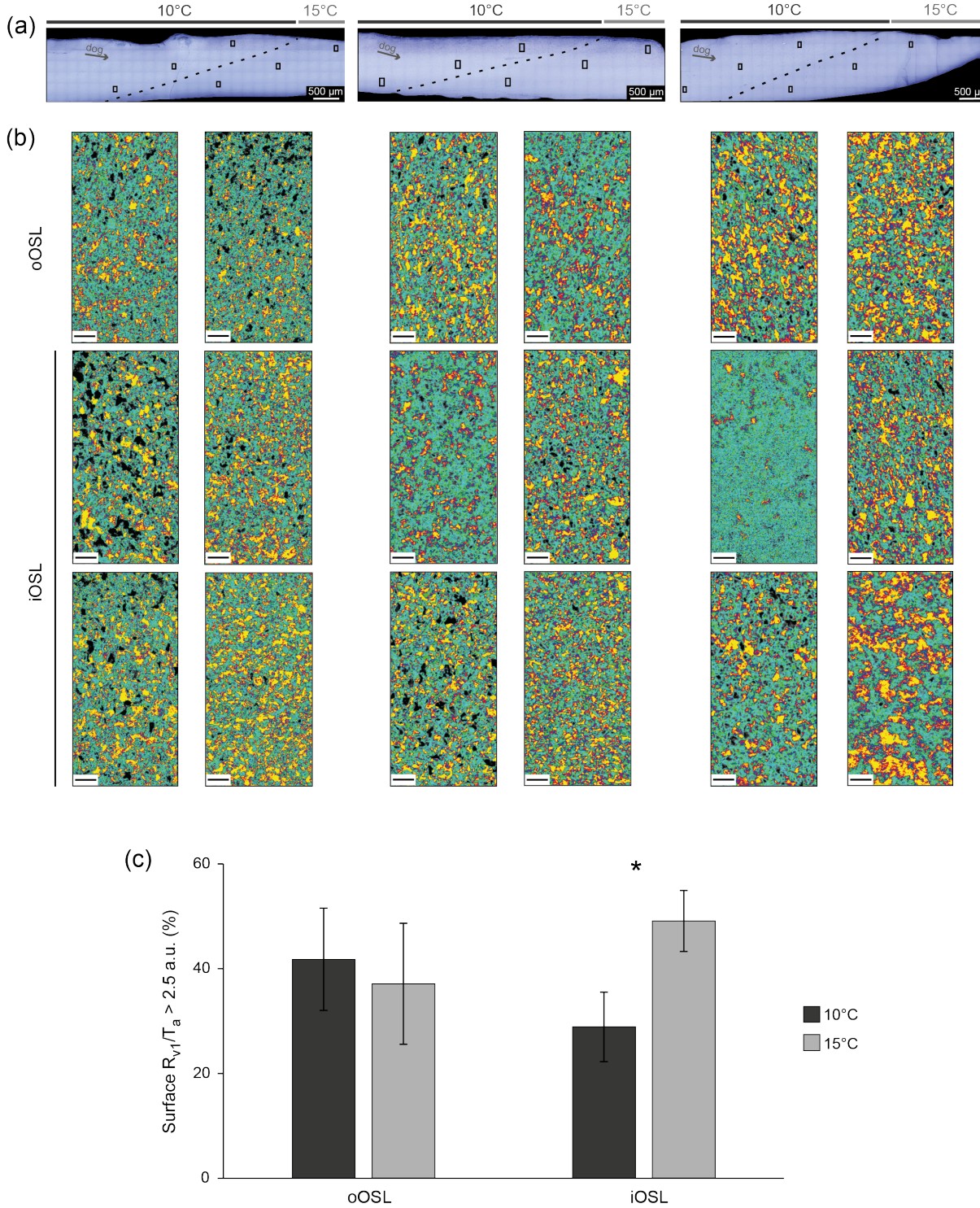

**Fig. 5.** Effect of temperature increase on biomineral orientation. (a) Position of the Raman maps of the three
specimens reared at 10 °C and 15 °C. Dotted lines indicate the location of the calcein marks. dog = direction
of growth. (b) Raman spectral maps of $R_{v1/Ta}$. Left images of each column represents shell portion formed
at 10 °C, right images represent shell portions formed at 15 °C. First row of pairs refers to oOSL, the other
two represent the iOSL. Scale bars = 10 µm. (c) Proportions of biominerals with $R_{v1/Ta} > 2.5$ a.u. with respect
to the total map area. Asterisks indicate significant difference between the orientation of iOSL
microstructures formed at 10 and 15°C ($p < 0.05$).


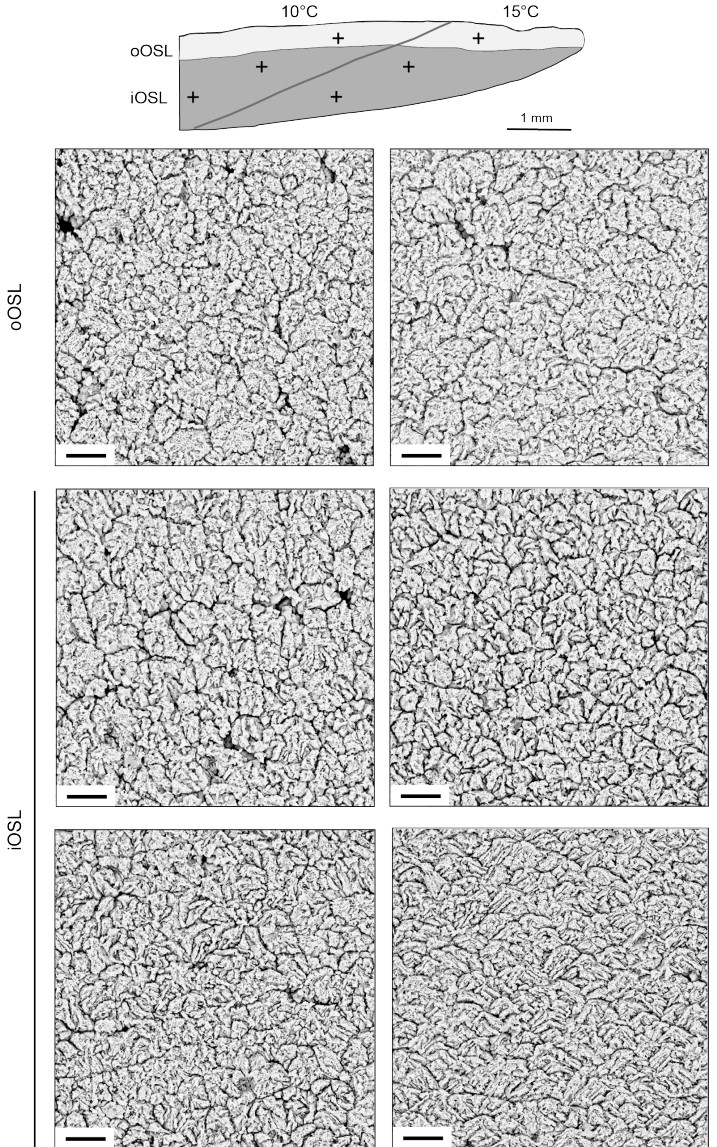



**Fig. 6.** SEM images of *Arctica islandica* shell microstructures formed at 10 °C (left column) and at 15 °C
(right column). The sketch indicates the position of the images 1 mm away from the calcein mark (grey
line).  The first row of images refers to the oOSL, the other two row refers to the iOSL. Scale bars if not
otherwise indicated = 5 μm.


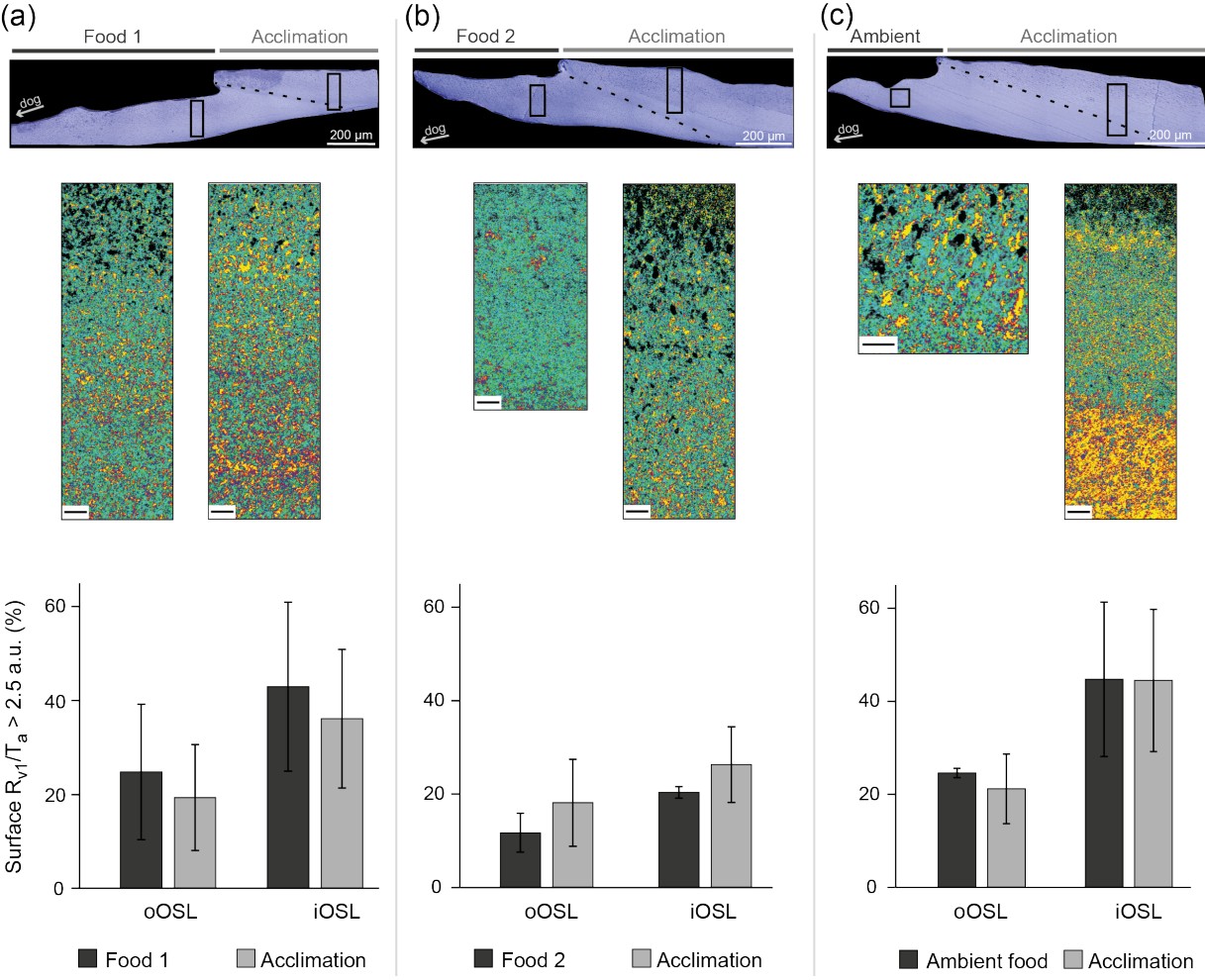

**Fig. 7.** Effect of different diets based on (a) food type 1, (b) food type 2 and (c) ambient food on biomineral orientation. The optical microscope images indicate the position of the Raman scans. Dotted line marks the start of the experiment. The portion of shell prior the line was formed during the acclimation phase. dog = direction of growth. The Raman spectral maps indicate the ratio $R_{v1/Ta}$ for each data point of the scan. For each shell, maps on the left represent shell portions during the experiment, maps on the right represent shell portions formed during the acclimation phase. In the acclimation portion of the sample reared with ambient food, a significant change in the microstructure orientation is visible. The respective area of the Raman map was not considered in further calculations because it was influenced by the emersion and transportation stress at the start of the experiment. Scale bars = 10 μm. The graphs show the proportions of biominerals of oOSL and iOSL with $R_{v1/Ta} > 2.5$ a.u. with respect to the total map area.



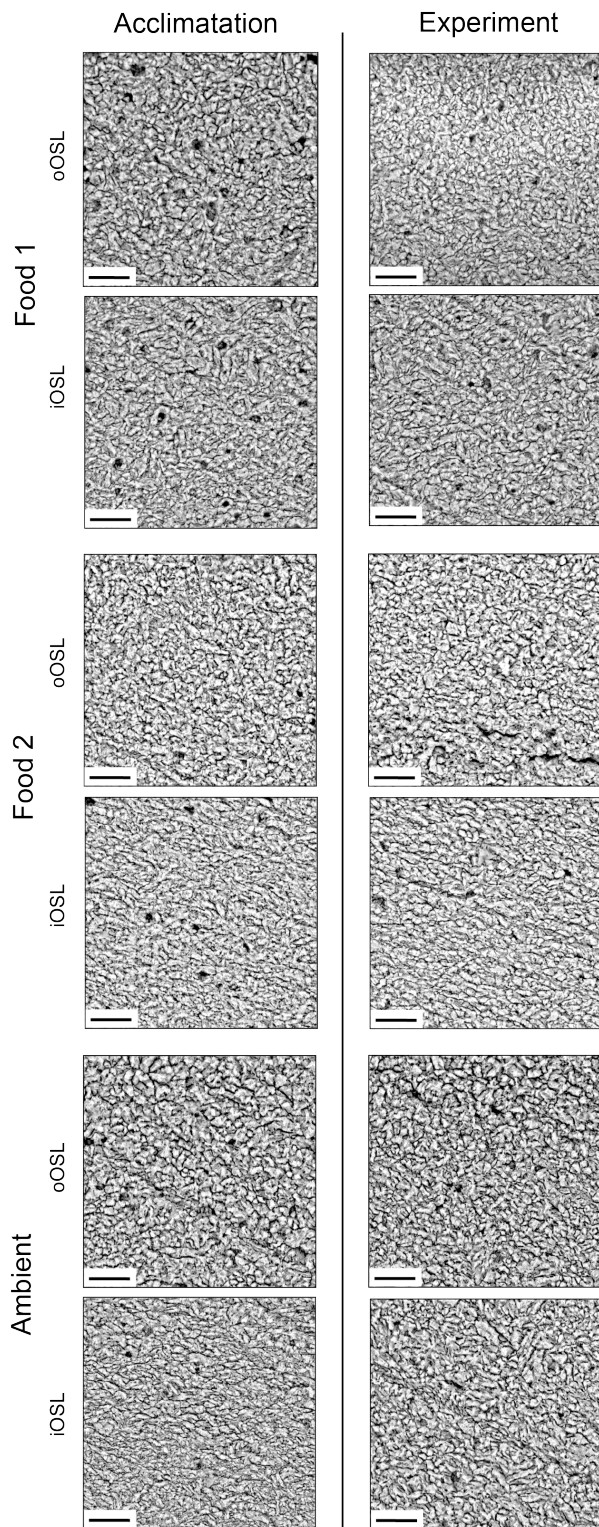


 **Fig. 8.** SEM images of *Arctica islandica* shell microstructures formed during the acclimation phase at AWI

(left column) and during the food experiment (right column). Scale bars = 4 µm.

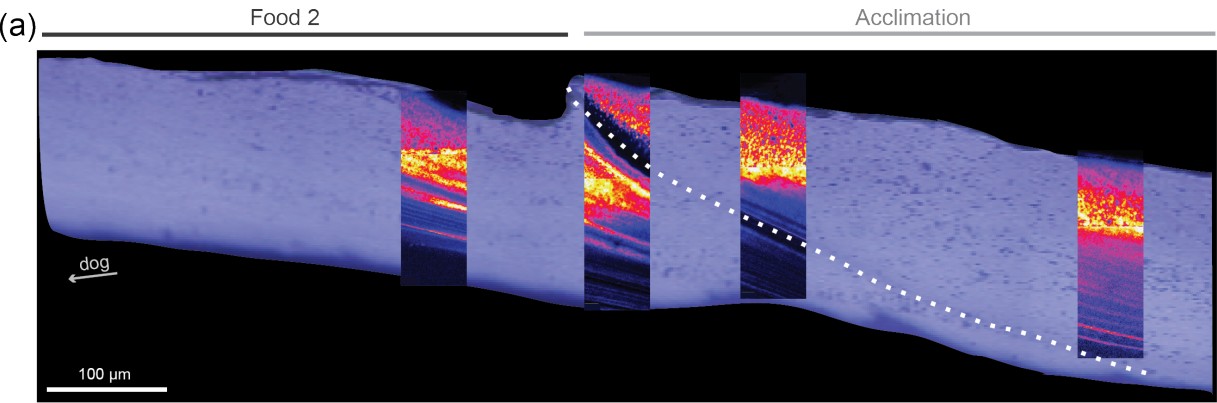

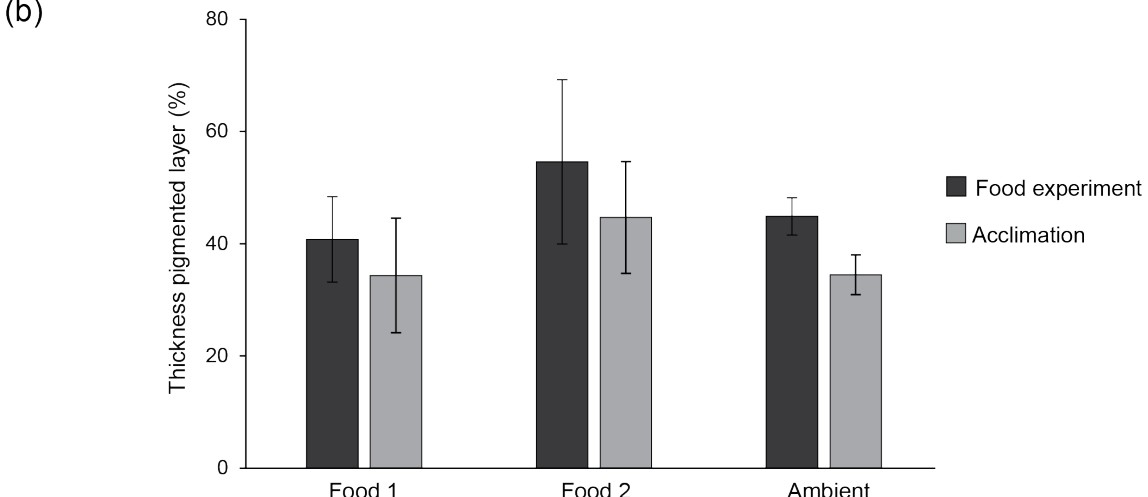


**Fig. 9.** Effects of diet on shell pigment distribution. (a) Raman spectral maps of the 1524 cm$^{-1}$ band
representing the distribution of the polyenes in the shell cultured with food type 2. Dotted line marks the
start of the experiment. dog = direction of growth. (b) The graph shows the thickness of the pigmented layer
over the whole shell thickness before and during the food experiments.


**Table 1.** List of the studied specimens of *Arctica islandica* and experimental conditions.

| Sample ID | Locality | Age | Experiment | Treatment |
|-----------|----------|-----|------------|-----------|
| A2 | Maine | 5 | Temperature | 10 °C + 15 °C |
| A4 | Maine | 4 | Temperature | 10 °C + 15 °C |
| A5 | Maine | 4 | Temperature | 10 °C + 15 °C |
| S12 | Kiel Bay | 1 | Diet | Food 1 |
| S14 | Kiel Bay | 1 | Diet | Food 1 |
| S15 | Kiel Bay | 1 | Diet | Food 1 |
| G11 | Kiel Bay | 1 | Diet | Food 2 |
| G12 | Kiel Bay | 1 | Diet | Food 2 |
| G15 | Kiel Bay | 1 | Diet | Food 2 |
| N13 | Kiel Bay | 1 | Diet | No additional food |
| N15 | Kiel Bay | 1 | Diet | No additional food |


**Table 2.** Details of the pigment composition of the *Arctica islandica* shells used in the food experiment.
The position of the major polyene peaks $R_1$ and $R_4$ in the Raman spectrum is indicated together with the
number of single and double carbon bonds of the pigment molecular chain ($N_1$ and $N_4$). Each shell was
analyzed in the portions formed before and during the experimental phase.

| Sample ID | Shell portion | $R_1$ (cm$^{-1}$) | $R_4$ (cm$^{-1}$) | $N_1$ | $N_4$ |
|-----------|---------------|--------------------|--------------------|-------|-------|
| S12 | Acclimation | 1130.9 | 1515.2 | 9.7 | 10.8 |
| | Food 1 | 1121.4 | 1515.3 | 12.1 | 10.7 |
| S14 | Acclimation | 1133.2 | 1519.4 | 9.3 | 10.2 |
| | Food 1 | 1132.2 | 1518.6 | 9.5 | 10.3 |
| S15 | Acclimation | 1129.5 | 1516.5 | 10.0 | 10.6 |
| | Food 1 | 1132.1 | 1519.8 | 9.5 | 10.1 |
| G11 | Acclimation | 1132.6 | 1518.4 | 9.4 | 10.3 |
| | Food 2 | 1129.5 | 1517.0 | 10.0 | 10.5 |
| G12 | Acclimation | 1131.7 | 1518.7 | 9.6 | 10.3 |
| | Food 2 | 1132.1 | 1518.2 | 9.5 | 10.4 |
| G15 | Acclimation | 1132.4 | 1519.5 | 9.4 | 10.2 |
| | Food 2 | 1128.0 | 1520.9 | 10.3 | 10.0 |
| N13 | Acclimation | 1130.2 | 1515.6 | 9.9 | 10.7 |
| | Ambient food | 1131.4 | 1514.1 | 9.6 | 10.9 |
| N15 | Acclimation | 1117.9 | 1516.0 | 13.3 | 10.6 |
| | Ambient food | 1130.7 | 1517.0 | 9.8 | 10.5 |
| Average | | 1129.7 ± 4.2 | 1517.5 ± 2.0 | 10.1 ± 1.1 | 10.4 ± 0.3 |