# Peer review of "The effects of environment on Arctica islandica shell formation and"

_Biogeosciences, 2016_

## Referee Comment (RC1) · Anonymous Referee #2 · 9 Jan 2017

Dear authors,

This manuscript represent a substantial amount of work, is scientifically grounded and in the scope of BGD. Given several minor corrections and/or improvements, it should in my opinion meet the criteria for publication in BG.

I have, In particular, two remarks : - A detailed description of the orientation of microstructural units in the different layers of A. Islandica shell is clearly missing. Please add a proper microstructural description, be it in the form of a sketch, a figure, a dedicated paragraph, etc.. Also, where are these units located/oriented in the SEM or CRM figure (for example, in figure 5) ? - The comparison with EBSD is maybe not so objective. . . ? CRM definitely has its advantages, but so does EBSD (mapping of the actual orientations of the crystallographic axis of the crystal lattice). Please consider

removing or rephrasing paragraph 4.2.

In a more general view, I regret that more environmental conditions (salinity, pH) were not tester when the authors had both the specimens and the required setup in place (adding one or two step to the initial temperature experiment wouldn't be so much hard work, compared to the time and effort the authors did put into testing the impact of feeding... ?). This would have allowed to estimate the real usefulness of crystalline orientation as a potential independent temperature proxy.

---

## Author Comment (AC1) · 20 Jan 2017

Dear Referee #2,

We appreciate your positive review and the constructive comments.

In regards to the first remark: to our knowledge a detailed description of the orientation of the microstructural units in the different shell layers of Arctica islandica currently not available. We are only aware of one single paper that has been published on a similar subject (Karney et al., 2012). However, in that case, the EBSD analyses were exclusively conducted in the hinge plate and not in the ventral margin. The study characterized the microstructural orientation in the growth increments and in the growth lines, respectively. However, no data was shown on the three shell layers mentioned in the current manuscript. Given the frequent use of A. islandica in sclerochronological

studies, the community will benefit from a map of crystal orientation in the whole shell, not just in very small shell portions.

As suggested, a more detailed description of the microstructures will be added to a revised version of the manuscript (paragraph 2.4: A. islandica shell organization) and a sketch (new Fig. 2) will be provided to better locate each type of microstructure described in the article. However, a superimposition of SEM images and CRM spectral maps would be imprecise since the two analyses were conducted on different machines without common coordinates as reference. We therefore prefer to show the two outputs separately.

The paragraph 4.2 will be edited as suggested. The main difference between the two techniques is the output of absolute (EBSD) and relative (CRM) data. This can be considered the main advantage of EBSD over the CRM. The information has been added to the paragraph. However, it should be realized that the "absolute orientation" is determined for the actual cross section. Therefore we regard a relative change in orientation as sufficient to for the questions asked in this study.

As for the third point highlighted by the referee, we agree that an experimental setup in which more parameters had been varied would be very interesting. However, this was outside the scope of the present study, which focused on the general feasibility of the applied methods. The positive results obtained by this first study set the basis for further studies in which now a more complex matrix of parameters can be investigated.

REFERENCES:

Karney, G.B., Butler, P.G., Speller, S., Scourse, J.D., Richardson, C.A., Schröder, M., Hughes, G.M., Czernuszka, J.T., Grovenor, C.R.M., 2012. Characterizing the microstructure of Arctica islandica shells using NanoSIMS and EBSD. Geochemistry, Geophys. Geosystems 13, doi:10.1029/2011GC003961

---

## Referee Comment (RC2) · Anonymous Referee #1 · 16 Feb 2017

In the paper "The effects of environment on Arctica islandica shell formation and architecture" Milano and co-authors investigate the structural properties of A. islandica shells as new environmental proxies. Despite its potential for future (palaeo-)oceanographic studies, the orientation of microstructural units in relation to environmental changes in shells has not yet received much attention. Therefore I consider this paper of high scientific quality and importance. The usage of the English language is excellent and the paper should definitely be published within BG.

However, in the following, I have some minor remarks on the draft, which should be addressed before publishing:

For the food experiments both the food and the temperature setting/conditions have been recorded. For the temperature experiment however, to my understanding, only

the temperature has been measured. It is stated that throughout the experiment the shells were fed by/with ambient water. Please clarify what this mean exactly. What if during the first half of the experiment there was an algal bloom, ie plenty of food? Would you have known? Can this possibility really be ruled out with this setup? It would have been good to keep a control group at 10°C for the second half of the experiment to rule out any influence of the food availability.

Line 345: "Altered crystallographic organization may derive from the animal exposure to suboptimal conditions." In this study different temperature settings and food compositions have been chosen but I would like to know a bit more about the "why". Why exactly have those settings been chosen? What temperature does A. islandica tolerate? When would we expect the animals to be stressed? Why 10° and 15°C?

Pigment distribution is mentioned throughout the manuscript and has been measured. I would like to know why. Why are those pigments important? What are they good for within the shells. Why bother about them? I am sure there are good reasons for that. Please clarify early in the text.

Line 299 "All treatments showed a slightly thicker pigmented layer formed during the experiment than during the acclimation phase": Is there an (hypothetical) explanation for it given later on in the discussion? Why is that? Maybe I missed it. If so, disregard this comment please.

Line 802: How and where exactly has the pigment thickness been measured? And what does it tell us? Does it occur in identical parts of the shell in different ontogenetic years?

For the water temperature experiment it seems that the shells have been kept in a lab for about 1.5 years after collection. What happened to the shells during that time? If it was other experiments, how certain is it that they still "behaved naturally"?

The transfer of the shells from AWI to NIOZ happened in January, if I read that correctly.

[Figure]

It is said that this transport created a stress line. However, January is usually not part of the growing season in A. islandica. How can the authors be certain that this line was built at that time/during the transport?

Line 287 "the shell portion deposited before the experiment": what time period is that exactly here? Acclimation? What happened to the shell at that time? How long before the experiment? "Before the experiment" does mean it has been under controlled conditions? Please clarify.

Line 326 and following: "As a consequence..." is that really true? I honestly don't know this myself but thinking of Tridacna and Spondylus with thick shells for warm water and Serripes with thin shells for cold water I am a bit in doubt if this is really such a general trend. Maybe it is the phrasing? It sounds to me as if that is always the case. Maybe relativize?

Line 754-756: why do polyene peaks occur exactly where aragonite peaks are? Has this to do with the aragonitic crystal lattice?

TECHNICAL CORRECTIONS Line 38: maybe consider changing "shells" to specimens? It is living animals and not just the shells.

Line 39 and following: no space between number and unit ($°C$).

Line 89: $\delta$18Oshell should be followed by "value" (or "ratio"?) and not stand alone.

Line 94: "cite Schone et al 2010 papers here" clearly has not been done yet.

Line 202: "Polarized Raman microscopy"? "Confocal" maybe?

Line 314: "pigmentationreact" two words.

Line 352: "Hedegaard et al., (2006)" no comma here.

Line 359: "specie-specific" here an "s" is missing.

Line 389 and following: "...and many other bivalves - is linked to environmental variables (e.g., Witbaard et al., 1997, 1999; Schöne et al., 2004; Butler et al., 2010; Mette et al., 2016)." Citations are all on A. islandica. What about "many other bivalves"?

Line 434: "sewater" here an "a" is missing.

Line 750: adult shell? It was 4 or 5 years old. That is not adult for A. islandica, or is it?

Line 751: "where" must be "were".

---

## Author Response (AR1)

Dear Referee #1,

The authors thank you for your thoughtful comments and remarks.

• In regards to the food supply during the temperature experiment, the clams only received food
from the outside ambient environment as seawater was pumped into the flow-through laboratory
at the Darling Marine Center (Walpole Maine, USA). We were able to verify that Chlorophyll
(pigment concentrations mg/m$^3$) remained fairly constant over the experiment (see screenshot
below). We used data from the Perry phytoplankton & optics lab
(http://perrylab.umeoce.maine.edu/docksampling.php), which is from the Darling Marine
Center. From these data, we do not think it is likely that food availability was substantially
different between the two culture periods.

• *A. islandica* optimal temperature range is between 6 and 10°C (Mann, 1982). The effect of water
temperature on the shell growth in this species is described in section 4.3. Based on the studies
cited, temperatures around 16 °C induce stress in the animals. In the current experiment, the
temperatures selected are within the natural range for this region (4-20 °C; Beirne et al., 2012
and screen shot below). Therefore, they are representative for water conditions to which the
species is regularly exposed and potentially influenced in the formation of its carbonate
structure.

[Figure]

• The specific functions of the shell pigments have not been disclosed yet. However, Stemmer
and Nehrke (2014) observed an enrichment in polyenes in the growth lines potentially
suggesting their involvement in the biomineralization process. Furthermore, the high phenotypic variation in pigmentation among and within mollusk species, indicates that these molecules may
not have a primary function as adaptive tools (i.e. camouflage, warning signaling) as in other
animals (Seilacher, 1972; Evans et al., 2009). This, in turn, can indicate a certain degree of
influence of the environment on the pigments, in particular by diet (Hedegaard et al., 2006;
Soldatov et al., 2013). In the current study, the effect of different dietary regimes was tested in
order to explore the potential of polyenes as environmental proxy. This information was added
to the section 2.4.

• The hypothetical explanation for a slightly thicker pigmented layer during the experimental
phase can be found at the end of section 4.1. There, we discussed the possibility that pigments
may be assimilated and therefore influenced by diet. However, the thickening that we observed
was not large enough to anyhow be supported by this hypothesis.

• The methodology used for measuring the pigmented layer thickness was added to section 2.5:
"*The images were analyzed using the software Panopea (© Schöne and Peinl). The thickness of*
*the pigmented layer was calculated as distance between the outer shell margin and the point*
*where the concentration of polyenes suddenly declined. The measurements were taken*
*perpendicular to the shell outer margin.*" The shells analyzed (food experiment) were all 1-
year-old. Therefore, the effect of ontogeny could not be tested. For this purpose, we would
suggest to select older specimen (> 20 years).

• The clams were collected in November 2009. They were then transported to the Darling Marine
center where they were kept in flow-through conditions in sediment similar to their natural
setting (fully described in Beirne et al., 2012). Again, these clams did not receive any "extra"
food and their environmental conditions were not manipulated. They were exposed to the
ambient conditions (temperature, salinity, and food) until this experiment began. Only then,
were water temperatures manipulated. Again, salinity and food conditions were identical to
those in the natural environment at the Darling Marine Center. We are confident that this
transplantation in 2009 did not impact the findings noted in the manuscript.

• In regards to the growing season of *A. islandica,* there are discrepancies in the literature
suggesting that the concept of a winter growth shutdown cannot be set in stone. Specimens from
Iceland indicate a growth slowdown from October to January (Schöne et al., 2005; Marali and
Schöne, 2015). However, *A. islandica* from Western Gulf of Maine display continuous growth
throughout the year (Wanamaker et al., 2008). In good agreement with these results, Beirne et
al. (2012) recorded growth during the winter months (January-March). Based on these
observations, we can conclude that more studies are needed to fully understand the growing
period of this species. However, we can already appreciate a certain degree of spatial variability.
Likely, populations from different geographic locations are influenced by specific conditions
which induce different growth responses. In addition, when talking about growing season, we
must consider that ontogeny plays a very important role. It is well known that juvenile/young shells grow more and for more extended periods of time compared to older specimens. In our
work, we focus on juveniles from the Baltic Sea. In the light of the previous observations, this
condition suggests an extended growing season throughout the year. As a consequence, it is
likely that a stress line can be formed in January. In support to this hypothesis, shell length was
measured at the beginning and at the end of the experiment. These data provided the exact
amount of carbonate produced over the experimental phase. By using this measurements, we
could locate the last shell portion formed before the experiment. In this spot all specimens
presented a clearly visible line, likely related to the transposition stress. No other growth lines
could be identified on the shells, supporting again the idea of a continuous growth.

• The shell portion mentioned as "deposited before the experiment,, in section 3.3 refers to the
shell deposition during the acclimation phase. To make the text clearer, the sentence was
rephrased. Section 2.2 describes the details of the food experiment. It is stated that the
acclimation period lasted for 3 weeks before starting the experiment (therefore, 18 Jan-8 Feb.
2015). Generally, the acclimation phase in scientific experiments represents that period of time
in which the animals are kept under the same conditions as during the experimental phase with
the exception of the investigated variable(s), in this case food.

• The sentence starting with "As a consequence..." refer to the thickness of the calcitic layer. Not
to the overall thickness of the shell. *Tridacna* and *Serripes* may not be the suitable species to
take into consideration, given that their shells are fully aragonitic. As for *Spondylus*, the shell is
composed by aragonite and calcite. The relative abundance of the single polymorphs support
our statement and the cited work by Lowenstam (1954) and Taylor and Kennedy (1969).  In
fact, as a warm-water species, the aragonite layer is much thicker that the calcitic one (Maier
and Titschack, 2010). Based on this data, we can conclude that the trend is indeed very well
represented in numerous mollusk species.

• The polyenes peaks do not occur at the same wavelengths as the aragonite peaks. As explained
in section 2.5, the two major polyene peaks are found at ~ 1130 ($R_1$) and 1520 $cm^{-1}$ ($R_4$). On the
other hand, the typical aragonite spectrum is characterized by peaks at 152, 206, 705 and
1085$cm^{-1}$.

**Technical corrections**

• Line 38: "Shells,, was changed into "specimens,, as suggested.
• Although the absence of strict rules, the space between values and units is commonly used and
accepted.
• Line 89: The word "value,, was added to "$\delta^{18}O_{shell}$,,.
• The missing references at line 94 were added.
• Line 202:  in the literature cited the method was referred as polarized Raman microscopy.
• Line 314: the typo was corrected.
• Line 352: the typo was corrected.

•  Line 359: the typo was corrected.

•  Line 389: "and many other bivalves,, was deleted.

•  Line 434: the typo was corrected.

•  Line 750: the distinction between adult/juvenile is not strictly fixed. Witbaaard et al. (1997)
define "juveniles,, shells with a height between 10 and 23 mm. The shells of the food experiment
were in the same range (10-14 mm) so they were defined as juveniles. The shells from the
temperature experiment measured between 38 and 44 mm. Based on the size classes of the
specimen analyzed in this work, they were categorized as juveniles and adults, respectively.
This distinction was also made to reinforce clarity in the text.

•  Line 751: the typo was corrected.

2012). However, in that case, the EBSD analyses were exclusively conducted in the hinge plate
and not in the ventral margin. The study characterized the microstructural orientation in the growth
increments and in the growth lines, respectively. However, no data was shown on the three shell
layers mentioned in the current manuscript. Given the frequent use of *A. islandica* in
sclerochronological studies, the community will benefit from a map of crystal orientation in the
whole shell, not just in very small shell portions. The used of CRM is therefore clearly
advantageous.

As suggested, a more detailed description of the microstructures will be added to a revised version
of the manuscript (paragraph 2.4: *A. islandica* shell organization) and a sketch (new Fig. 2) will be
provided to better locate each type of microstructure described in the article. However, a
superimposition of SEM images and CRM spectral maps would be imprecise since the two analyses
were conducted on different machines without common coordinates as reference. We therefore
prefer to show the two outputs separately.

The paragraph 4.2 will be edited as suggested. The main difference between the two techniques is
the output of absolute (EBSD) and relative (CRM) data. This can be considered the main advantage
of EBSD over the CRM. The information has been added to the paragraph. However, it should be
realized that the "absolute orientation" is determined for the actual cross section. Therefore we
regard a relative change in orientation as sufficient to for the questions asked in this study.

As for the third point highlighted by the referee, we agree that an experimental setup in which more
parameters had been varied would be very interesting. However, this was outside the scope of the
present study, which focused on the general feasibility of the applied methods. The positive results
obtained by this first study set the basis for further studies in which now a more complex matrix of
parameters can be investigated.

**References**

[revised manuscript text omitted]